# Isolated Intramural Hematoma of Superior Mesenteric Artery: Case Reports and a Review of Literature

**DOI:** 10.3390/diagnostics13233581

**Published:** 2023-12-01

**Authors:** Marta Ascione, Rocco Cangiano, Alireza Mohseni, Andrea Molinari, Antonio Marzano, Alessia Di Girolamo, Luca Di Marzo, Wassim Mansour

**Affiliations:** 1Vascular and Endovascular Surgery Division, Department of General Surgery and Surgical Specialties, Policlinico Umberto I, “Sapienza” University of Rome, Viale del Policlinico, 155, 00161 Rome, Italy; rocco.cangiano@uniroma1.it (R.C.); andrea.molinari@uniroma1.it (A.M.); antonio.marzano@uniroma1.it (A.M.); alessia.digirolamo@uniroma1.it (A.D.G.); luca.dimarzo@uniroma1.it (L.D.M.); 2Faculty of Medicine and Surgery, “Sapienza” University of Rome, Viale Regina Elena 324, 00161 Rome, Italy; mohseni.1974637@studenti.uniroma1.it

**Keywords:** intramural hematoma, superior mesenteric artery, treatment, steroid

## Abstract

(1) Background: Spontaneous isolated intramural hematoma of the superior mesenteric artery (SIHSMA) is a rare entity often considered as a subset of spontaneous isolated dissection of the superior mesenteric artery (SIDSMA). It is characterized by a completely thrombosed false lumen with or without an ulcer-like projection with computed tomography (CT) imaging. The recent literature describes few reports with a relatively short-term follow-up. The natural course, prognosis, and treatment options for SIHSMA still lack consensus. We present two cases of acute abdominal pain in a young man due to IMH of the superior mesenteric artery with an extensive literature review. (2) Case report: A 46-year-old male patient was submitted to an urgent CTA for acute abdominal pain, showing the presence of an isolated dissection of the superior mesenteric artery, determining significant stenosis of the vessel with collateral vessel patency. The patient referred to a recent COVID-19 infection, whose course was paucisymptomatic. He was conservatively treated with antiplatelet therapy and corticosteroid treatments, and, after a few days, the symptomatology completely regressed; also, the 2-month-control CTA showed complete IMH regression and the absence of any signs of residual stenosis. The second patient was a 61-year-old male patient who was submitted to an urgent CTA for acute abdominal pain, showing the presence of an isolated dissection of the superior mesenteric artery, not determining significant vessel stenosis. He was conservatively treated with antiplatelet therapy and corticosteroid treatment, and after a few days, the symptomatology completely regressed and the radiological control showed complete dissection regression. (3) Conclusion: SISHSMA is a rare entity of vascular pathology, and conservative management represents the best medical strategy. We propose corticosteroid treatment as one of the most appropriate tools in the conservative treatment of SISHSMA.

## 1. Introduction

Spontaneous isolated intramural hematoma of the superior mesenteric artery (SIHSMA) represents a sporadic event, often considered as a subset of spontaneous isolated dissection of the superior mesenteric artery (SIDSMA) [1]. IMH was first described by Krukenberg in 1920 as a “dissection without intimal tear”. It is defined as clotted blood within the artery wall (tunica media), in the absence of an identifiable continuum with the arterial lumen, as an intimal tear. The IMH has historically been considered a ruptured vasa vasorum in the arterial media; however, recent studies have shown that a small intimal tear could be detected. Thus, some considered IMH as a subset of spontaneous isolated dissection [2,3,4,5]. 

The natural course, prognosis, and treatment options for SIHSMA still lack consensus. We report a case of complete clinical and radiological regression, after a high dose of corticosteroid treatment. 

## 2. Case Report

### 2.1. Case A

A 46-year-old male patient with a history of controlled moderate hypertension was admitted to the emergency department for severe sudden-onset epigastric pain. The patient was apyretic during his stay. The patient referred to a recent COVID-19 infection, whose course was paucisymptomatic. No history of trauma was found. The abdomen was tractable and not painful or sore on superficial and deep palpation. Peristalsis was present. Blood examination revealed elevated inflammatory markers (increased C-reactive protein and white blood cell count, without procalcitonin elevation). Blood cultures and urine cultures were negative. Inflammatory, immunological, and genetic analysis panels revealed an increase in inflammatory index without a genetic or immunological correlation. A thorax–abdomen–pelvis CT angiography (Figure 1) exam revealed the presence of a subintimal eccentric parietal thickening, neatly hyperdense, extended along the entire course of the SMA vessel and more prominent in the proximal section, approximately 1 cm from the origin, not determining a significant lumen stenosis (Figure 1a). The imaging confirmed the diagnosis of intramural hematoma of the superior mesenteric artery (SIHSMA). Since the patient was hemodynamically stable, conservative medical therapy was started based on antiplatelet therapy, antalgic drugs, and blood pressure control. According to the systemic inflammatory condition and in the absence of clinical or laboratory signs of infection, starting a high dose of corticosteroid treatment was deemed necessary, beginning with 25 mg of prednisone twice daily. Soon after therapy started, the symptoms resolved and, after 2 months of conservative treatment with corticosteroids, the control CTA (Figure 1b,d) revealed the SMA patency with complete intramural hematoma regression in the absence of residual stenosis.

### 2.2. Case B

A 61-year-old male patient was submitted to an urgent CTA (Figure 2) for acute abdominal pain, showing the presence of an isolated SIHSMA not determining significant vessel stenosis. The patient reported no fluctuations in his symptoms and was stable. Abdomen physical examination was devoid of signs. Past medical history did not reveal any immunological or rheumatological disease. The inflammatory panel showed a systemic inflammation with C-reactive protein elevation and increased white blood cell count. No signs of infectious disease were found, considering procalcitonin levels and the negative results of the blood cultures, urine tests, and cultures. Considering the case’s complexity, a conservative management approach consisting of antiplatelet therapy and a high dose of corticosteroid treatment was opted for. After a few days, the symptomatology completely resolved, and the radiological follow-up performed after two months showed complete IMH regression (Figure 2b,d).

## 3. Material and Methods

### 3.1. Literature Review

A systematic literature search and critical appraisal of the collected studies were conducted according to the Preferred Reporting Items for Systematic Review (PRISMA) standards. An electronic search was performed on PubMed from December 2022 to April 2023. The search terms were (“intramural hematoma of the superior mesenteric artery” OR “intramural hematoma of visceral artery“ OR “intramural hematoma of digestive artery” OR “ intramural hematoma of aorta“ OR “acute pulmonary haemorrhage” OR “acute pulmonary haemorrhage”) AND (“idiopathic” OR “isolated” OR “acute”) AND (“treatment” OR “corticosteroid treatment” OR “conservative treatment” OR “management” OR “immunological disease” OR “infective disease” “genetic disease” “autoimmune correlation”). The bibliographies of all identified articles were reviewed and cross-referenced to search for additional relevant literature. Methodological evaluation of each study was conducted according to PRISMA standards, including bias assessment. 

Data collection involved study selection and data extraction. Articles with titles or abstracts relevant to our review were analyzed, and those studying the clinical presentations, disease management, outcomes, and pharmacological choices in conservative treatment were selected. A consensus process resolved disagreements on eligibility between researchers. 

Unpublished, pre-print, or non-English language literature was excluded.

Other exclusion criteria were publications dated prior to 2000, no apparent correlation with the topic sought, generalization of IMH to non-visceral digestive districts (brain, upper or lower limbs), or districts of major interest (heart, aortic arch, ascending aorta, descending thoracic aorta, abdominal aorta). Data extraction was performed by two researchers and verified by two others. 

### 3.2. Results

A review of the titles and abstracts and a manual search of the reference lists were carried out. The reference lists of all identified articles were reviewed to find missed-out papers. This search identified 48 articles screened after reading the full articles. Thirty-two articles were discarded as unrelated to the topic. The resulting 16 articles were screened to exclude duplicates and irrelevant papers. Four non-English papers and five papers published before 2000 were discarded after reading and analysis based on the exclusion criteria. 

The following inclusion criteria were used: (1) observational and retrospective and multicentric studies, (2) reviews and mini-reviews, and (3) case reports/series. These publications were carefully evaluated, considering the review’s main aims. This evaluation left seven scientific papers comprising one observational study, three retrospective studies, and three case reports. 

Of the seven articles, only two specifically address SIHSMA. Three papers discuss the dissection of the SMA, with intramural hematoma described as a potential subset. Another paper presents a case report on intramural hematoma of the gastric artery, while the final article details a case report of multiple dissections and visceral intramural hematoma. The search strategy is depicted in Figure 3 for clarity.

Fifty-five cases of superior mesenteric artery intramural hematoma were found in the explored database. The main characteristics of the articles included in this review are summarized in Table 1. The studied population was composed of 10 females and 45 males. The main symptoms were gastrointestinal symptoms such as abdominal pain (83.63%), and few asymptomatic cases were found (16%). The main risk factors were recorded as the same in the vascular population: hypertension (19 patients), diabetes mellitus (2), smoke (8), coronary artery disease (5). No reference to genetic disease (Marfan syndrome or Ehlers–Danlos syndrome) or immunological disease was found. Laboratory analysis and instrumental analysis were not available in almost all cases.

Computed tomography angiography (CTA) was performed in all cases where instrumental data were reported. The IMH was described as the presence of a circular or crescent-shaped thickening of the aortic wall of >5 mm in the absence of detectable blood flow. It is classified according to their size and position along the SMA and the follow-up frequencies (Table 2). 

The screening of the updated literature revealed the effectiveness of conservative treatment in 76 percent of the patients. The progression of intramural hematoma to dissection or aneurysm was found in 5 patients (9%), with one case needing endovascular treatment after 1.5 months.

Six patients underwent endovascular treatment during the hospitalization for persistent abdominal pain symptoms, with only one needing open surgery.

There was no mention of using steroids or other therapeutics (as increasing antiplatelet therapy) as part of the conservative management, even in patients whose pain symptoms persisted (Table 3).

## 4. Discussion

Spontaneous isolated intramural hematoma of the superior mesenteric artery (SIHSMA) is a rare, complex pathology to clinically define and identify. 

The literature of the last 50 years discusses SIHSMA in relation to the isolated dissection of the superior mesenteric artery (SIDSMA). Only 27 cases of SIDSMA [7] are described with an estimated incidence of less than 0.06% [6]. Nevertheless, the superior mesenteric artery (SMA) is the most common incident site of isolated dissection among all digestive visceral arteries. SIDSMA predominantly affects the male population (ratio 3:1) and the sixth–seventh decade of age (between 60 and 70 years) [8].

With the increased use of CTA, which has enhanced the diagnosis of spontaneous isolated IMH and dissection, there has been a rise in the number of studies published concerning the management and clinical outcomes of both SISMAD and SIHSMA [1,8,9,10,11]. 

Only two recent Asian studies [1,8] had specifically investigated the SIHSMA both observationally and retrospectively but with a relatively short-term follow-up.

Some different authors such as Li [9], Sakamoto [12], Yun [13], and Yoo [14] have suggested their classification of SIHSMA using similar criteria, such as the false lumens’ partial or complete thrombosis, the presence of penetrating ulcers, and true lumen occlusion. However, there is still no standardized classification for SIHSMA. 

The etiology of SIHSMA is still unclear; however, the presence of vascular wall inflammation has been hypothesized by some authors [3,15,16] as the leading cause of intramural hematoma. There are multiple associations between IMH and artery dissection along with risk factors such as hypertension, arteriosclerosis, smoke, dyslipidemia, genetic diseases (Marfan’s disease, Elhers–Danlos disease), and systemic infections (mainly viral) as shown by the study of Nienaber from 2003 [5]. Hypertension is considered the primary risk factor for IMH and dissection; in fact, it directly causes parietal stress on the vessel wall and indirectly activates proinflammatory agents during macrophage recruitment. 

Based on the literature, intramural hematoma is recognized as a form of vascular wall inflammation. Given that the systemic inflammatory profile serves as a predictive factor for SIHSMA (akin to acute aortic syndrome), exploring all potential causes of systemic inflammatory disease is essential. This encompasses the examination of immunological, genetic, and infectious correlations to elucidate the multifactorial events in SISHMA.

The immunology department was consulted to examine all possible immunological profiles. Initially, the focus was on autoantibodies associated with the coagulation system, including cardiolipin antibodies, beta-2 glycoprotein 1 antibodies, Antiphospholipid Antibodies (APAs), and Lupus Anticoagulants (LAs). Subsequently, investigations were conducted for systemic autoimmune diseases using autoantibody tests such as antinuclear antibodies (ANAs), extractable nuclear antigens (ENAs), Anti-Neutrophil Cytoplasmic Antibodies (ANCAs), Anti-Double-Stranded DNA (anti-dsDNA), Anticentromere Antibodies (ACAs), antihistone antibodies, Cyclic Citrullinated Peptide Antibodies (CCP), extractable nuclear antigen antibodies (e.g., anti-SS-A (Ro) and anti-SS-B (La), anti-RNP, anti-Jo-1, anti-Sm, Scl-70), and Rheumatoid Factor (RF). All immunological tests, specific for coagulation and systemic inflammation, yielded negative results. 

Genetic profiles were also examined, focusing on prevalent genetic pathologies in arterial diseases, including Marfan syndrome, Ehlers–Danlos disease, and Loeys–Dietz syndrome. In these instances, all tests yielded negative results. Consequently, no correlations between genetic and immunological factors were identified in any of the cases. In evaluating the inflammatory condition in SIHSMA patients as a potential infectious etiology, it was noted that none of the patients presented with pyrexia during their hospital stay. Comprehensive investigations were undertaken, encompassing blood cultures, urine tests, urine cultures, and a series of primary viral tests. These tests included immunoglobulin blood evaluations for varicella-zoster virus, measles, rubella, herpes simplex virus 1, herpes simplex virus 2, and cytomegalovirus. Notably, all these investigations returned negative results. 

In the first patient, SARS-CoV-2 was considered as a potential primary inflammatory factor. Several authors have described the pathogenesis of coagulopathy associated with COVID-19 (CAC) as a predominant cause of COVID-19-related vasculopathy [17,18,19,20,21,22]. In this instance, the conventional risk factor, combined with systemic inflammation attributed to the SARS-CoV-2 virus, might provide a foundation for the proliferation of wall inflammation in the superior mesenteric artery. In subsequent cases, no discernible association was observed with their systemic inflammatory status.

Conservative management was opted for following the guidelines and classifying our patients as uncomplicated cases. 

The ESVS [23] for IMH and SIDSMA and the ACC/AHA guidelines [24] for aortic suggest conservative management (Ia in ESVS; Ib-NR in ACC/AHA) in uncomplicated cases, with pressure control, and anticoagulant/antiplatelet and analgesic drugs [11,25]. Hemodynamically unstable patients or those presenting with complicated IMH/dissection need endovascular or open surgery vascular treatment (IIa ESVS). In addition, the studies of Xiao et al. [6] and Wang et al. [1], which represent the most recent and complete studies on SIHSMA, suggest conservative management in hemodynamically stable patients with no progression of IMH in 87% of cases [1] and complete remodeling in 62,5% of patients [6]. Antiplatelet and anticoagulant drugs, analgesics, and pressure control as conservative management were used. On the other hand, hemodynamically unstable patients presenting with complicated IMH or dissection with the persistence of abdominal pain with or without other clear clinical and radiological signs of acute mesenteric ischemia, require endovascular or open surgical vascular treatment (IIa ESVS). However, if acute mesenteric ischemia is doubtful, a preventive laparoscopy might be a good clinical practice to finalize the endovascular treatment of SIHSMA.

As Xiao and Wang’s studies have shown, seven patients (out of 55 cases) indicated immediate surgical intervention due to hemodynamically unstable conditions. The six-month CTA control depicted the progression of IMH to dissection or aneurysm in four patients, with just one case indicating an endovascular approach after 1.5 months since the beginning of symptoms, for exacerbation of abdominal pain and clear CTA evidence of a penetrating ulcer.

In this context, a conservative strategy was employed among the non-complicated patients identified in this study. This involved arterial pressure stabilization, pain management, and antiplatelet and anticoagulant therapy standardization. However, one patient from this cohort remained unresponsive to analgesic therapy and continued to report pain, necessitating the exploration of an alternative strategy. 

In order to consider the inflammatory condition as the primary cause of the SIHSMA etiopathology, anti-inflammatory treatments are ideal therapeutic agents. Furthermore, this massive systemic inflammation was devoid of immunological, genetic, or infectious associations based on our previously debated laboratory studies. Thus, the corticosteroid drugs were suggested as the main anti-inflammatory agent. Considering SISHMA a systemic inflammation and categorizing the two patients from this study as uncomplicated, the administration of corticosteroid drugs was deemed suitable as a conservative management approach.

The optimal dosage and specific corticosteroids were selected in consultation with the immunology department for maximal efficacy against systemic inflammatory disease. A regimen of 25 mg of prednisone was administered twice daily for a minimum of two months. Subsequently, the symptoms in the initial patient subsided.

Two months post initiation of the treatment, a follow-up CTA was conducted and revealed complete resolution of the SIHSMA in both patients, with no residual vessel stenosis. Consequently, the decision was made to discontinue the therapy. Furthermore, a six-month follow-up CTA was performed indicating the long-term effects of steroids, confirming the complete resolution of the IMH without vessel remodeling or other enhancement differences after stopping conservative steroid treatment. Contrary to the guidelines recommending conservative treatment and suggesting potential IMH regression six months post-symptom onset, the case report presented here demonstrates complete IMH reabsorption just two months after and no dissection and/or aneurysm progression after six months of symptom onset through CTA control, following the initiation of conservative treatment with corticosteroid drugs. Furthermore, as discussed in the Literature Review section, neither the specific SIHSMA studies by Xiao et al. [6] and Wang et al. [1], nor any papers and reports from recent literature on visceral IMH or aortic IMH [15], refs. [25,26,27,28,29,30,31,32,33,34,35,36,37,38,39,40,41,42,43,44,45] have documented the use of corticosteroids as a primary conservative treatment for intramural hematoma. This study represents the first instance of corticosteroids being considered an effective treatment for IMH of the superior mesenteric artery, deemed an “off-label treatment”, with complete regression observed after two months. However, the findings are constrained by a limited sample size and data from a single institution. Additionally, the observation period was insufficient to ascertain the potential progression of IMH requiring annual control in order to evaluate the impact of steroids on IMH, long-term effects, and possible complications.

## 5. Conclusions

This case highlights a rare vascular pathology, SIHSMA, characterized by ambiguous etiology and uncertain management strategies. Critical evaluation and conservative management are the optimal medical strategies in hemodynamically stable patients. In this study, it is demonstrated that in uncomplicated cases, devoid of any infective, immunological, or genetic correlations, a high dose of corticosteroid treatment can be considered an ‘off-label treatment’ and a viable option in the conservative treatment arsenal for SIHSMA. However, it is imperative to note that the relationship between IMH and corticosteroid treatment remains elusive, primarily due to the limited number of cases in the present study. Consequently, further scientific investigations, taking into account the patient’s medical history, are indispensable to formulate the most effective medical treatment strategy for SIHSMA.

## Figures and Tables

**Figure 1 diagnostics-13-03581-f001:**
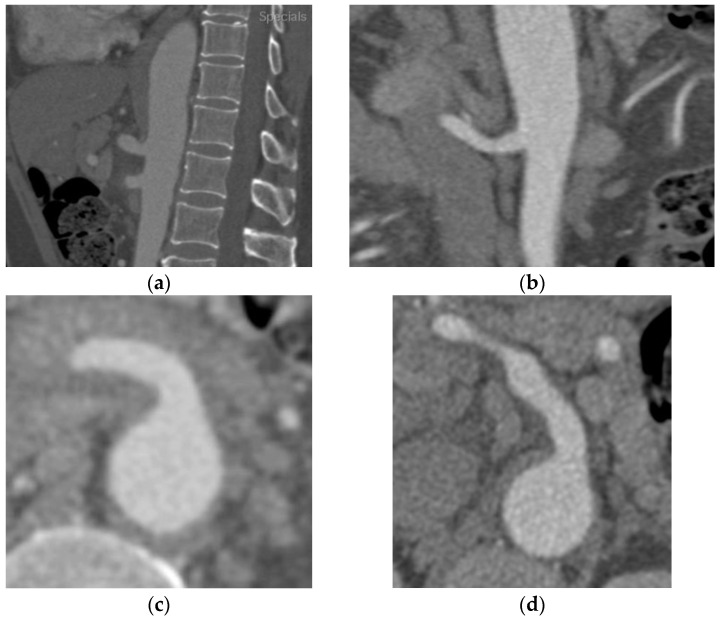
Thorax–abdomen–pelvis computed tomography angiography (CTA) showing (**a**) in the sagittal section and (**c**) in the transverse section a hypodense area around the superior mesenteric artery determining significant stenosis at the origin of the vessel. (**b**) The two-month-control CTA after corticosteroid treatment revealed a complete absorption of the edema and the vessel’s patency without any stenosis visible in the sagittal section, (**d**) as well as clearly in the transverse section.

**Figure 2 diagnostics-13-03581-f002:**
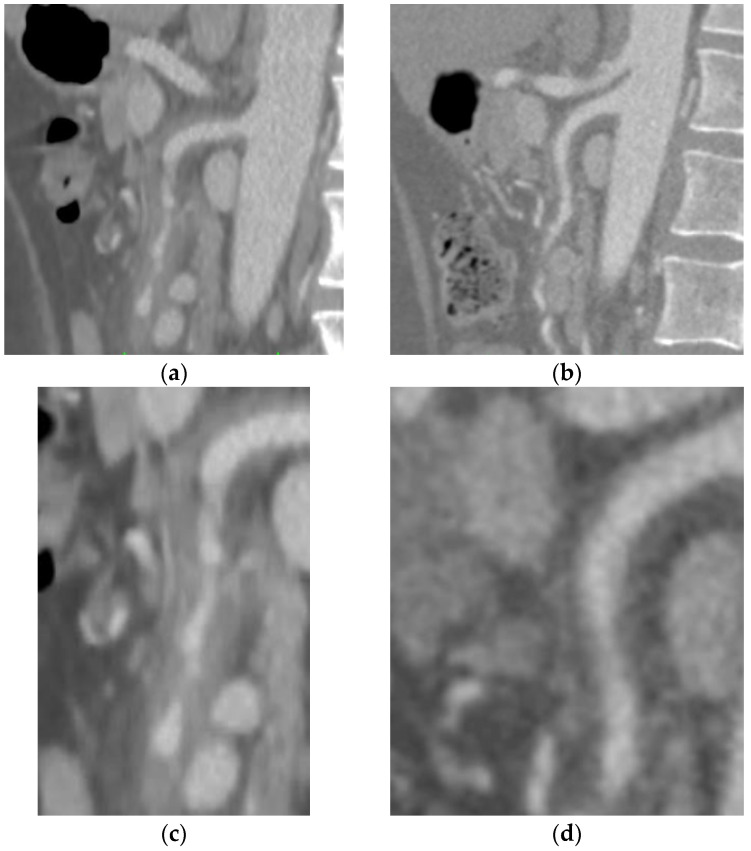
(**a**) Emergency thorax–abdomen–pelvis computed tomography angiography (sagittal section) showing subintimal eccentric parietal thickening extended along the entire course of the SMA vessel, not determining significant stenosis of the vessel (**c**) with caliber alterations leading to rosary chain stenosis. (**b**) The two-month-control CTA after corticosteroid treatment revealed complete absorption of the hematoma and a complete patency of the vessel (**d**) without any sign of vessel interruption.

**Figure 3 diagnostics-13-03581-f003:**
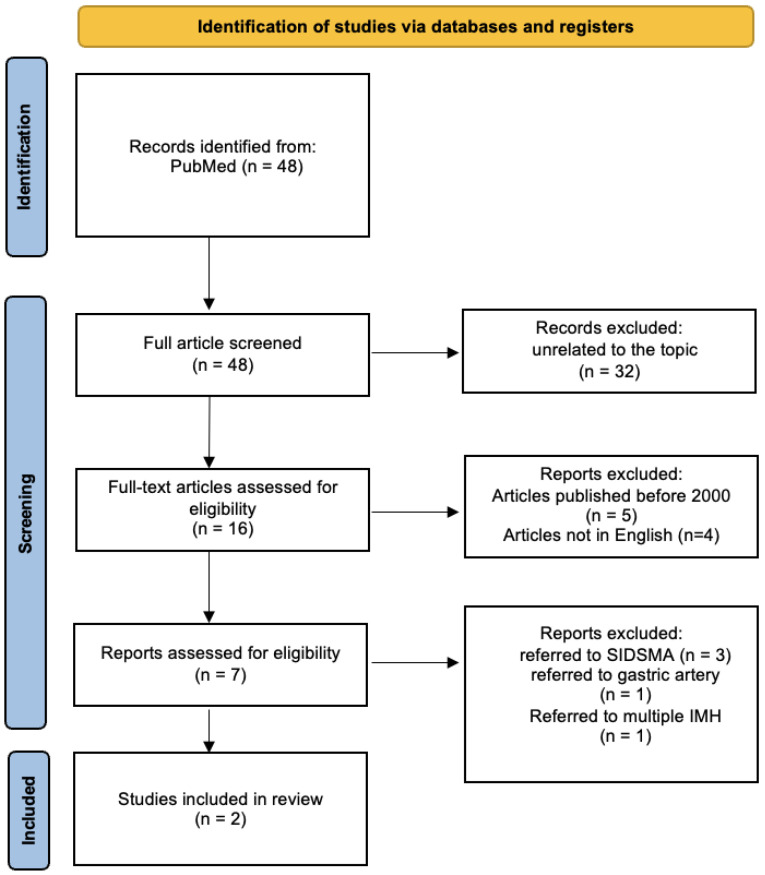
Search strategy: a methodological appraisal of each study was conducted according to the PRISMA standards, including bias assessment. The data collection process included study selection and data extraction. (SIDSMA: spontaneous isolated dissection of superior mesenteric artery).

**Table 1 diagnostics-13-03581-t001:** The table shows a summary of the findings of all reported clinical cases of intramural hematoma of the superior mesenteric artery (SISHMA) in the literature. (M, male; F, female).

*N*. (Patient)	Reference	Type of Article	Onset Symptom	Risk of Factors
31 (M/24; F/7)	Wang, Yingliang et al. [1]	Observational study	- 6 (6/31, 19%) asymptomatic with incidental diagnosis - 25 (25/31, 81%) gastrointestinal symptoms	Hypertension, 11 (35%) Atherosclerosis, 6 (19%) Diabetes mellitus, 2 (6%) Smoking, 5 (16%) Valvular heart disease, 3 (10%) Coronary arterial disease, 4 (13%) Hyperlipidemia, 2 (6)
24 (M/21; F/3)	Xiaoq, Z et al. [6]	Retrospective study	- 21/24 Gastrointestinal symptoms (abdominal pain) - 3/24 asymptomatic (occasional occurrence)	Hypertension, 8 (32%) Coronary artery disease, 1 (4%) History of abdominal surgery, 3 (12%) Current smoker, 3 (12%)

**Table 2 diagnostics-13-03581-t002:** Data collection of laboratory and imaging findings of SIHSMA from a systematic review of the current literature. (N/A, not available; CTA, computed tomography angiography; IMH, intramural hematoma).

Reference	Laboratory Data	Instrumental Analysis	Other
Wang, Yingliang et al. [1]	N/A	Computed tomography angiography (CTA): - 21 (21/31, 68%) cases with the IMH involving the branches of the SMA (ileocolic or distal ileal arteries) - 10 (10/31, 32%) cases with the IMH limiting to the trunk of SMA.	CTA or ultrasound images as follow-up at 1 month, 3 months, 6 months, and 12 months and every year thereafter
Xiaoq, Z et al. [6]	N/A	Computed tomography angiography (CTA): arteries were measured and classified according to anatomic descriptors	CTA control: 7 days after the initial diagnosis, and 1 month and 6 months after admission.

**Table 3 diagnostics-13-03581-t003:** Data collection of treatment management and outcomes from a systematic review of current literature. Also underlining the use of corticosteroid as a possible management treatment. (IMH, intramural hematoma).

Reference	Treatment Management	Outcomes	Use of Corticosteroid
Wang, Yingliang et al. [1]	- 28 patients (26/31, 87%) were conservatively treated (fasting, pain management, blood pressure control, vasodilator drugs, anticoagulation drugs, anti-aggregant drugs). - 3 patients (3/31, 9.67%) underwent endovascular stenting for persistent abdominal pain symptoms	- 1/31 progression of intramural hematoma to penetrating ulcer at CT and underwent stent implantation and coil embolization 1.5 months later due to persistent abdominal pain - 1/31 progression from hematoma to dissection 7 months later but remained stable. - 27/31 no progression of hematoma - All the patients who had stent implantation (*n* = 3) maintained a patent vascular lumen and complete absorption of IMH.	NO
Xiaoq, Z et al. [6]	- 20\24 conservative treatment using fasting, pressure control, vasodilators, antiaggregants, anticoagulants (LMWH—enoxaparin; warfarin; rivaroxaban) * - 1\24: Exploratory laparotomy + embolectomy + arteriotomy. - 3\24: visceral vessel stenting **	- 15/24 (62.5%) Complete remodeling - 4/24 (16.67%) Partial remodeling - 2/24 (8.33%) No change - 3/24 (12.5%) Dissection remodeling or aneurysm change - 22 patients (91.6%) showed angiographic improvement to complete remodeling.	NO

* Conservative treatment duration: between 3 and 5 days; treatment with anticoagulants for 3 to 6 months. ** Endovascular stent placement was deemed necessary if pain symptoms were resistant to pain-relieving treatment.

## Data Availability

Data sharing is not applicable; no new data were created or analyzed in this study.

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
