# Peer review of "Isolated Intramural Hematoma of Superior Mesenteric Artery: Case Reports and a Review of Literature"

_diagnostics, 2023, doi:10.3390/diagnostics13233581_

Round 1

Reviewer 1 Report

Comments and Suggestions for Authors

As the condition is rare, but very clearly presernted I sggest to publish the paper in presernt stater

Author Response

Dear Reviewer, we sincerely appreciate the time and effort you have dedicated to reviewing our manuscript. Your insightful comments and suggestions have been invaluable in enhancing the quality and clarity of our work. We are deeply grateful for the opportunity to address your feedback and improve our submission. Below, we present our responses to each of your comments :

Thank you for your encouraging remarks regarding the presentation of our paper. We are honored by your suggestion to publish the paper in its current state and are motivated by your recognition of our work on this rare condition. Your positive feedback is greatly appreciated and has been a source of inspiration for our team.

Once again, we extend our heartfelt thanks to each reviewer for their thorough reviews and valuable contributions to our work. Your expertise and guidance have been instrumental in refining our manuscript. We look forward to our revised manuscript being favorably considered for publication.

Warm regards,

Authors

--------------------------------------------------

Reviewer 2 Report

Comments and Suggestions for Authors

My congrats for your paper:

*concise and focused

*well writted and structured

*figures are informative

Reviews of such an infrequent findings, especially with some variants of teh already available cases determined by frank dissection, woudl be always awaited and useful.

Comments on the Quality of English Language

Adequate.

Author Response

Dear Reviewer, we sincerely appreciate the time and effort you have dedicated to reviewing our manuscript. Your insightful comments and suggestions have been invaluable in enhancing the quality and clarity of our work. We are deeply grateful for the opportunity to address your feedback and improve our submission. Below, we present our responses to each of your comments :

We are deeply grateful for your positive assessment of our paper, particularly your comments on its conciseness, structure, and the informative nature of the figures. Your commendation is not only a source of encouragement but also a valuable affirmation of our approach to presenting infrequent findings in the field. Thank you for your kind words and constructive insights.

Once again, we extend our heartfelt thanks to each reviewer for their thorough reviews and valuable contributions to our work. Your expertise and guidance have been instrumental in refining our manuscript. We look forward to our revised manuscript being favorably considered for publication.

Warm regards,

Authors

--------------------------------------------------

Reviewer 3 Report

Comments and Suggestions for Authors

The authors presented two cases of spontaneous intramural hematoma of superior mesenteric artery. The article is very interesting and I enjoyed reading it. 

- does the response to corticosteroids indicate an autoimmune etiology? Did you investigate for vascualitis?

- the prisma chart is the old version

- It could be better if you started with the systematic review and included your cases and compare them with others form the literature. the current format is unusual

-  

Comments on the Quality of English Language

Minor changes 

Author Response

Dear Reviewer, we sincerely appreciate the time and effort you have dedicated to reviewing our manuscript. Your insightful comments and suggestions have been invaluable in enhancing the quality and clarity of our work.

We are deeply thankful for your valuable feedback and for your appreciation of our article's contribution to the field. Your inquiry regarding the autoimmune etiology and the response to corticosteroids is insightful and has indeed been a significant aspect of our research. As elaborated in the discussion section, we have thoroughly investigated the possibility of vasculitis, including the assessment of Anti-neutrophil Cytoplasmic Antibodies (ANCA), both c-ANCA and p-ANCA, to comprehensively address the pathology.

Regarding the PRISMA chart, we would like to clarify that we utilized the most current version available at the time of our submission, sourced directly from the PRISMA website (2020 version).

We understand the importance of using up-to-date resources and have ensured that our manuscript reflects the latest standards and guidelines. We also acknowledge your comments on the structure of our article. The journal's categorization of our work as a case report necessitates adherence to a specific format. This format includes integrating the literature review within the materials and methods section, resulting in the unique structure you noted. We appreciate your understanding of these constraints and have endeavored to present our findings as clearly and comprehensively as possible within the given framework.

Once again, we extend our heartfelt thanks to each reviewer for their thorough reviews and valuable contributions to our work. Your expertise and guidance have been instrumental in refining our manuscript. We look forward to our revised manuscript being favorably considered for publication.

Warm regards,

Authors

--------------------------------------------------

Reviewer 4 Report

Comments and Suggestions for Authors

Authors presented a rare entity review: spontaneous isolated intramural hematoma of the superior mesenteric artery. I consider the topic original and relevant; moreover it address a specific gap in vascular pathology treatment
This paper adds an holistic vision of vascular complications and introduces corticosteroid treatment as one of the most appropriate tools in the conservative treatment. Further controls should be considered a multicentre trial
Conclusions are consistent with the evidence and arguments presented
and they address the main question posed
References, tables and figures are appropriate?

Author Response

Dear Reviewer, we sincerely appreciate the time and effort you have dedicated to reviewing our manuscript. Your insightful comments and suggestions have been invaluable in enhancing the quality and clarity of our work. 

Thank you for your encouraging and thoughtful evaluation of our work. We are particularly grateful for your recognition of the originality and relevance of our topic in addressing a specific gap in vascular pathology treatment. Your comments on the holistic vision of vascular complications and the introduction of corticosteroid treatment as a key tool in conservative treatment are highly valued.

We concur with your suggestion regarding the necessity of further controls and the potential for a multicentre trial. This is an aspect we will certainly consider for future research. In response to your query about the references, tables, and figures, we have taken great care to ensure that they are not only appropriate but also central to illustrating the efficiency of steroids and summarizing the relevant literature. These elements are fundamental to a comprehensive understanding of this rare pathology, its various etiologies, and its epidemiology worldwide.

Once again, we extend our heartfelt thanks to each reviewer for their thorough reviews and valuable contributions to our work. Your expertise and guidance have been instrumental in refining our manuscript. We look forward to our revised manuscript being favorably considered for publication.

Warm regards,

Authors